# Expression Profiles of Genes Related to Serotonergic Synaptic Function in Hypothalamus of Hypertensive and Normotensive Rats in Basal and Stressful Conditions

**DOI:** 10.3390/ijms26157058

**Published:** 2025-07-22

**Authors:** Olga E. Redina, Marina A. Ryazanova, Dmitry Yu. Oshchepkov, Yulia V. Makovka, Arcady L. Markel

**Affiliations:** 1Federal Research Center Institute of Cytology and Genetics, Siberian Branch of Russian Academy of Sciences (SB RAS), Novosibirsk 630090, Russia; oredina@bionet.nsc.ru (O.E.R.); ryazanova@bionet.nsc.ru (M.A.R.); makovkayv@bionet.nsc.ru (Y.V.M.); 2Department of Natural Sciences, Novosibirsk State University, Novosibirsk 630090, Russia

**Keywords:** hypothalamus, short-term restraint stress, serotonergic synapse, gene expression, hypertension, RNA-Seq, ISIAH rat strain

## Abstract

The hypothalamus belongs to the central brain structure designed for the neuroendocrine regulation of many organismal functions, including the stress response, cardiovascular system, and blood pressure, and it is well known that the serotonergic hypothalamic system plays a significant role in these processes. Unfortunately, the genetic determination of serotonergic hypothalamic mechanisms has been little studied. The aim of this article is to describe the expression profile of the genes in the hypothalamic serotonergic synapses in hypertensive ISIAH rats in comparison with normotensive WAG rats in control conditions and under the influence of a single short-term restraint stress. It was found that 14 differentially expressed genes (DEGs) may provide the inter-strain differences in the serotonergic synaptic function in the hypothalamus between the hyper- and normotensive rats studied. In hypertensive rats, downregulation of *Slc18a1* gene in the presynaptic serotoninergic ends and decreased expression of *Cacna1s* and *Htr3a* genes determining the postsynaptic membrane conductance may be considered as a main factors causing differences in the function of hypothalamic serotoninergic synapses in hypertensive ISIAH and normotensive WAG rats at the basal conditions. Under basal conditions, glial cell genes were not involved in the formation of inter-strain differences in serotonergic synaptic function. The analysis of transcriptional responses to restraint stress revealed key genes whose expression is involved in the regulation of serotonergic signaling, and a cascade of interrelated changes in biological processes and metabolic pathways. Stress-dependent changes in the expression of some DEGs are similar in the hypothalamus of hypertensive and normotensive rats, but the expression of a number of genes changes in a strain-specific manner. The results suggest that in hypothalamic glial cells of both strains, restraint stress induces changes in the expression of DEGs associated with the synthesis of Ip3 and its receptors. Many of the identified serotonergic DEGs participate in the regulation of not only serotonergic synapses but may also be involved in the regulation of cholinergic, GABAergic, glutamatergic, and dopaminergic synapses. The results of the study provide new information on the genetic mechanisms of inter-strain differences in the functioning of the hypothalamic serotonergic system in hypertensive ISIAH and normotensive WAG rats at rest and under the influence of a single short-term restraint (emotional) stress.

## 1. Introduction

As Stephanie W. Watts [1] aptly noted, “Since its discovery … serotonin has been an elusive candidate as a substance that plays a role in the disease of … hypertension”. Indeed, to this day, one can find many conflicting opinions on this issue. The organismal regulatory effects of serotonin are so numerous and in some cases diverse that it is often impossible to rely upon the primary or indirect ways of its influence on the state of actual hemodynamics and arterial pressure levels. The results of different studies sometimes do not coincide and sometimes even contradict each other, which is most likely due to the multi-target action of serotonin and the structural, functional, and localization diversity of its receptors. Most serotonin is synthesized in the enterochromaffin cells of the gastrointestinal tract, and a significantly smaller, but no less important, proportion of serotonin is synthesized in various parts of the brain, mainly in the raphe nuclei. Neurons in these nuclei send many ascending and descending fibers. Ascending axonal fibers from the rostral and superior group of raphe nuclei reach the hypothalamus [2,3]. However, serotonin can also be synthesized in the hypothalamic nuclei. It was found that perikarya of some neurons in the nucleus arcuatus and in the dorsomedial nuclei and the median eminence also contain 5-HT [2,4]. The increase in serotonin concentration in these neurons after local injections of tryptophan confirms the ability of these hypothalamic neurons to synthesize serotonin. The data on serotonin synthesis in the hypothalamus were supported by immunohistochemistry [5] and by in vivo autoradiography analyses [6]. As for serotonin receptors, a high concentration of 5-HT2c receptor-like immunoreactivity and some amount of 5-HT2a-positive fibers were detected in the suprachiasmatic nucleus [7]. The 5-HT(1A) receptor immunoreactivity was observed in several hypothalamic nuclei [8]. Thus, the hypothalamic serotonin system, in addition to the close cooperation with serotonin and other neurotransmitters from different brain structures, may play the role of an autonomous regulator of hypothalamic neurotransmission and influence the regulation of basic physiological processes. However, the features of the functioning of serotonergic synapses of the hypothalamus in normotensive and hypertensive subjects, as well as the molecular events occurring in serotonergic synapses under stress conditions, have been little studied.

There are several strains of hypertensive rats that serve as useful models for studying the mechanisms of development of various forms of hypertension. One of them is the ISIAH rat strain. ISIAH rats (inherited stress-induced arterial hypertension), selected for a sharp increase in systolic blood pressure under short-term (30 min) restraint stress [9,10], are a model of a stress-sensitive form of hypertension with genetically pre-determined activation of the hypothalamic–pituitary–adrenal (HPA) and sympathetic adrenal systems. The neuroendocrine changes observed in their various physiological systems under stress are similar to those observed under emotional stress in humans [11].

Previously, by studying the dynamics of the activation of early response genes in the hypothalamus of ISIAH rats exposed to a single short-term restraint stress, we showed that the upregulation of the *Fos* gene, which is a marker of neuronal activation, coincides with the dynamics of a stress-induced increase in arterial blood pressure in ISIAH rats. Maximum activation of the *Fos* gene transcription was observed 1 h after the onset of stress, and it remained at the same level for the next hour [12]. A study of the hypothalamic transcriptomes of hypertensive ISIAH and normotensive WAG rats demonstrated that two hours after the onset of this type of stress, a change in the transcription level of numerous genes that form both the common responses to stress and the responses to stress characteristic of hypertensive and normotensive rats (strain-specific responses to stress) is observed in the hypothalamus of both ISIAH and WAG rats. General characteristics of transcriptomes were published in [13,14]. These reports considered the main biological processes associated with the response to stress (including the response to oxidative stress), signaling, and ion transport. However, within the framework of these publications, it was not possible to consider in detail the molecular genetic mechanisms of synaptic transmission characteristic of certain types of synapses. Since this area is poorly studied, it deserves detailed consideration. In connection with a special invitation from the editors of the IJMS journal to participate in the special issue “Serotonin in Health and Diseases”, the authors performed a comparative description of the molecular genetic mechanisms of serotonergic synaptic transmission in the hypothalamus of hypertensive and normotensive rats at rest and under stress. Thus, the aim of this article is to describe the expression profile of the genes of the hypothalamic serotonergic synapse in hypertensive ISIAH rats in comparison with normotensive WAG rats in control conditions and under the influence of a single short-term (2 h) restraint stress.

## 2. Results

Transcriptomic analysis identified 3057 hypothalamic DEGs in an inter-strain (ISIAH/WAG) comparison at rest; 3603 genes altered their expression during restraint stress in the hypothalamus of hypertensive ISIAH rats (ISIAH DEGs), and 3157 genes altered their expression during restraint stress in WAG rats, as described in more detail in [14].

According to the KEGG database, 130 genes are assigned to the metabolic pathway associated with the serotonergic synapse. In our study, this list of genes was used to characterize, at the transcriptome level, the differences in the functional activity of serotonergic synapses in the hypothalamus of hypertensive ISIAH and normotensive WAG rats at rest and after exposure to a single short-term (2 h) restraint stress.

### 2.1. Inter-Strain Differences at Rest

#### 2.1.1. Characteristics of Differentially Expressed Genes

The analysis of inter-strain differences, without the influence of the stress factor, allowed the identification of 14 serotonergic differentially expressed genes (sDEGs) associated with the function of serotonergic synapse in the hypothalamus of hypertensive ISIAH and normotensive WAG rats. The list of these sDEGs includes four genes associated with hypertension and three genes for the sympathetic nervous system (Table 1).

Clustering of sDEGs is shown in Figure 1. Most of them (10 genes) have decreased transcription levels in the hypothalamus of hypertensive ISIAH rats. Some of the identified DEGs associated with serotonergic synapse function may control signaling using different neurotransmitters (Figure 2).

#### 2.1.2. Location of sDEGs Determining Inter-Strain Differences at Rest on the KEGG Map for Serotonergic Synapse

Using the KEGG database map, the positions of 14 sDEGs on the pre- and postsynaptic regions of the serotonergic synapse were determined (Figure 3). According to the positions of sDEGs on the KEGG map, inter-strain differences in the functioning of the presynaptic terminal are associated with the expression of the *Slc18a1* gene encoding the vesicular monoamine transporter, which acts to accumulate cytosolic monoamines into vesicles. Its expression is significantly reduced in ISIAH rats. The remaining DEGs are associated with postsynaptic signaling.

Inter-strain differences in the functioning of the postsynaptic membrane are associated with a decrease in the expression of the *Cacna1s* and *Htr3a* genes. *Cacna1s* enables voltage-gated calcium channel activity. The *Htr3a* gene belongs to the ligand-gated ion channel receptor superfamily. A decrease in *Htr3a* expression can lead to low membrane conductance and response amplitude. Thus, the decrease in *Slc18a1* expression in the presynaptic region and the decrease in the expression of the *Cacna1s* and *Htr3a* genes, which determine changes in the conductivity of the postsynaptic membrane, can probably be considered as the main factors determining the changes in postsynaptic signaling pathways in the serotonergic synapse associated with the remaining sDEGs listed in Table 1. According to the spatial distribution of sDEGs in the schematic representation of the serotonergic synapse (Figure 3), it can be concluded that glial cells are not involved in the formation of inter-strain differences in the functioning of the serotonergic synapse in the hypothalamus of hypertensive and normotensive rats at rest.

### 2.2. Stress Response

#### 2.2.1. Genes That Changed Transcription Levels When Exposed to a Single Short-Term Restraint Stress

In the hypothalamus of hypertensive ISIAH rats, the transcription level of 26 genes associated with the functioning of the serotonergic synapse changed upon exposure to stress. Most of these sDEGs (21 genes) were down-regulated by stress. In the hypothalamus of normotensive WAG rats, the response to stress was detected for 22 sDEGs, the majority (18 genes) of which decreased their transcription level in the hypothalamus of WAG rats upon stress. Clustering of the identified sDEGs associated with the response to a single short-term (2 h) restraint stress in the hypothalamus of hypertensive ISIAH and normotensive WAG rats is shown in Figure 4.

The results of functional annotation of sDEGs show that the genes involved in stress response in the hypothalamus of both rat strains affect numerous biological processes (Appendix A). The most enriched GO terms are associated with calcium ion transmembrane import into the cytosol, behavior, and synaptic signaling. The most enriched KEGG terms are associated with multiple signaling pathways (Appendix A). Functional analysis highlighted that in the hypothalamus of both rat strains, a large number of the analyzed genes are not specific for serotonergic synapses but may also control cholinergic, dopaminergic, glutamatergic, and GABAergic synapses (Appendix A). This non-specificity of a number of identified sDEGs suggests that they may play a significant role in modulating the functioning of not only serotonergic but also other synapses under stress.

Functional analysis of sDEGs suggested the presence of some similar links in the mechanisms of stress response in the hypothalamus of hypertensive and normotensive rats. Accordingly, we will further consider in more detail the common and strain-specific response to stress in the hypothalamus of ISIAH and WAG rats.

#### 2.2.2. Common Response to Stress

Comparison of the hypothalamic response to stress in hypertensive and normotensive rats revealed 12 common sDEGs (Figure 5). The transcription level of 11 of them decreases under stress. All 12 sDEGs were shown to change expression under stress in the same direction in the hypothalamus of both rat strains (Table 2). The list of these DEGs includes one gene associated with hypertension and four genes for the sympathetic nervous system.

#### 2.2.3. Location of Common sDEGs on the KEGG Map for Serotonergic Synapse

Changes in the functioning of the serotonergic presynaptic terminal and membrane in the hypothalamus of both rat strains under stress are associated with downregulation of the *Htr1a*, *Cacna1b*, and *Maoa* genes (Figure 6). According to the Harmonizome 3.0 database, all three genes are associated with sympathetic nervous system activity (Table 2).

Changes in the functioning of the postsynaptic membrane are associated with a decrease in the expression of the *Cacna1b*, *Cacna1d*, and *Htr1a* genes. Changes in the membrane potential of the postsynaptic cell lead to changes in signaling, which is realized through a decrease in the expression of *Gnaq* and *Gnb4*, and with the participation of the *Plcb1* and *Plcb2* genes, the expression of which also decreases during stress. It is important to note that, according to the KEGG scheme (Figure 6), glial cells are also involved in the stress response in the hypothalamus of both rat strains, which occurs through downregulation of *Gnaq*, as well as the *Plcb1* and *Plcb2* genes.

#### 2.2.4. Functional Annotation of Common DEGs

Functional annotation of common sDEGs shows that the most significantly enriched terms for biological processes are Serotonin metabolic process, Calcium ion import across plasma membrane, Regulation of neurotransmitter levels (Figure 7). The modulation of chemical synaptic transmission under stress observed in the serotonergic synapse of the hypothalamus of both rat strains affects numerous metabolic pathways, including the GnRH (gonadotropin-releasing hormone) signaling pathway, retrograde endocannabinoid signaling, cortisol synthesis and secretion, renin secretion, oxytocin signaling pathways, insulin secretion, calcium signaling pathways, thyroid hormone signaling pathways, relaxin signaling pathways, pshospholipase D signaling pathway, MAPK signaling pathways, Wnt signaling pathways, and cAMP signaling pathways (Figure 7). From Figure 7, it is clearly seen that sDEGs (*Cacna1b, Cacna1d*), whose function is associated with changes in the potential of pre- and postsynaptic membranes, as well as sDEGs (*Gnaq*, *Plcb1*, and *Plcb2*), associated with signaling both in the postsynaptic cell of the serotonergic synapse and in glial cells, play a key role in the regulation of these metabolic pathways.

#### 2.2.5. ISIAH-Specific Stress Response

The strain-specific response to stress in the hypothalamus of ISIAH rats is represented by a group of 14 sDEGs, the expression of 10 of which is reduced by restraint stress (Table 3). The group of genes associated with the specific response to stress in the hypothalamus of ISIAH rats includes six sDEGs associated with hypertension and one gene for the sympathetic nervous system.

#### 2.2.6. Location of ISIAH-Specific sDEGs on the KEGG Map for Serotonergic Synapse

Strain-specific changes in presynaptic membrane function under stress in the hypothalamus of ISIAH rats are associated with a decrease in the expression of the *Cacna1a* and *Htr1b* genes. Changes in postsynaptic membrane function under stress in the hypothalamus of ISIAH rats are associated with downregulation of the *Cacna1a*, *Cacna1c*, *Htr1b*, and *Kcnj6* genes. A decrease in the expression of these genes, which change the postsynaptic membrane potential, provides further signaling through a decrease in the expression of *Gnao1*. In addition, the *Itpr2* and *Itpr3* genes play an important role in postsynaptic signaling. The proteins encoded by these genes belong to the inositol 1,4,5-triphosphate receptor family (Ip3r), whose members mediate a level of cytoplasmic calcium through the production of inositol triphosphate. It should be noted that Ip3r controls the stress response not only in postsynaptic cells but also regulates calcium channel activity in glial cells (Figure 8).

#### 2.2.7. Functional Annotation of 14 sDEGs Associated with ISIAH-Specific Stress Response

Functional annotation of 14 sDEGs associated with ISIAH-specific stress response indicates a significant role of biological processes related to calcium ion transmembrane import into the cytosol and regulation of ion transport. According to the KEGG database, these sDEGs belong to a large number of signaling pathways: long-term depression, oxytocin signaling pathways, GnRH signaling pathways, relaxin signaling pathways, estrogen signaling pathways, cortisol synthesis and secretion, renin secretion, calcium signaling pathways, Ras signaling pathways, cGMP-PKG signaling pathways, phosphatidylinositol signaling system, MAPK signaling pathways, PI3K-Akt signaling pathways, and PI3K-Akt signaling pathways (Figure 9). It is evident from the figure that the regulation of most of these signaling pathways is associated with the expression of sDEGs *Cacna1c*, *Itpr2*, and *Itpr3*. Accordingly, it can be concluded that these genes may play a key role in regulating the functioning of the serotonergic synapse in the hypothalamus of hypertensive ISIAH rats under conditions of restraint stress.

#### 2.2.8. WAG-Specific Stress Response

The strain-specific response to stress in the hypothalamus of normotensive WAG rats is represented by a group of 10 sDEGs, the expression of 7 of which is reduced by restraint stress (Table 4). This group includes two genes associated with hypertension and one gene for the sympathetic nervous system.

#### 2.2.9. Location of WAG-Specific sDEGs on the KEGG Map for Serotonergic Synapse

Strain-specific changes in the presynaptic terminal of the serotonergic synapse are associated with a decrease in the *Maob* gene expression in the hypothalamus of WAG rats under stress conditions. On the membrane of the postsynaptic cells, changes are associated with a decrease in *Htr2a* gene expression. Further modulation of serotonergic synaptic transmission is associated with changes in the expression of *Gnai1*, *Plcb4*, and *Prkcg* genes. Changes in *Htr2a* expression can also alter glial cell function, which is associated with an increase in *Plcb4*, which catalyzes the formation of inositol 1,4,5-trisphosphate and diacylglycerol from phosphatidylinositol 4,5-bisphosphate (Figure 10).

#### 2.2.10. Functional Annotation of 10 sDEGs Associated with WAG-Specific Stress Response

Functional annotation of 10 sDEGs associated with WAG-specific stress response indicates a significant role of biological processes related to the positive regulation of ATP metabolic process, regulation of potassium ion transport, regulation of ion transport, and protein phosphorylation. According to the KEGG database, these DEGs belong to a large number of signaling pathways, the most enriched of which are long-term depression, long-term potentiation, oxytocin signaling pathways, GnRH signaling pathways, thyroid hormone signaling pathways, sphingolipid signaling pathways, relaxin signaling pathways, estrogen signaling pathways, VEGF signaling pathways, phospholipase D signaling pathways, chemokine signaling pathways, and calcium signaling pathways (Figure 11). The regulation of most of these signaling pathways is associated with the expression of sDEGs *Map2k1*, *Plcb4*, *Prkacb*, *Prkcg*, *Pla2g4a*, and *Gnai1*. Accordingly, it can be concluded that these genes may play a key role in regulating the functioning of the serotonergic synapse in the hypothalamus of normotensive WAG rats under restraint stress.

#### 2.2.11. Enrichment Analysis of Promoter Regions of Serotonergic DEGs

Enrichment analysis of promoter regions of serotonergic DEGs was performed using the Enrichr resource. The construction of the biomolecular interaction networks between transcription factors and serotonergic DEGs was performed in Cytoscape Web (Figure 12). Transcription factors that can control the common stress response, as well as strain-specific stress responses associated with the serotonergic synapse in the hypothalamus of ISIAH rats (Rarb, Egr1, and Rest) and WAG rats (Tfap2a) were identified.

## 3. Discussion

The functional role of serotonin as a neurotransmitter and physiological regulator is very broad; it plays a significant part in the regulation of behavior, neuroendocrine functions, the cardiovascular system, gastrointestinal homeostasis, etc. [15,16]. Unfortunately, data on the role of serotonin in the regulation of blood pressure and in the pathogenesis of hypertension remain unclear [17]. Peripheral serotonin controls many cardiovascular functions [18] but the data concerning the blood pressure regulation remains very contradictory [1,17]. As for the central serotonin regulation of the blood pressure, there is a higher degree of certainty since it is known that brain serotonin is involved in the regulation of sympathetic tone and the function of the neuroendocrine system that produces hormones directly addressed to the cardiovascular system, including stress hormones [19]. In the latter aspect, the role of the hypothalamus is especially important [20].

In the present study, we used previously obtained data [13,14] to conduct a comparative analysis of hypothalamic gene expression profiles associated with the functioning of the serotonergic synapses in hypertensive ISIAH and normotensive WAG rats. Inter-strain differences of gene expression at the control conditions and after a single short-term restraint (emotional) stress were evaluated. Below, we discuss the most interesting genes, from our point of view, whose functions are associated with the presynaptic neuronal terminal and presynaptic membrane, as well as with the postsynaptic membrane, since these sDEGs trigger activation of an ion flow, membrane depolarization, and a second-messenger signaling cascade in the postsynaptic cell.

Comparison of differences in the expression profiles of genes associated with the functioning of the serotonergic synapse in the hypothalamus of hypertensive ISIAH and normotensive WAG rats at rest showed that inter-strain differences in the presynaptic neuronal terminals are associated with a decrease in the expression of the *Slc18a1* gene encoding the vesicular monoamine transporter Vmat1 which acts to accumulate cytosolic monoamines (including serotonin) into vesicles [21]. Despite the fact that a population-based study using randomly selected polymorphisms in the *SLC18A1* gene showed its significant association with hypertension [22], the *VMAT1/SLC18A1* gene has not been associated with the development of hypertension to date. However, the missense variation Thr136Ile in the *VMAT1/SLC18A1* gene has been characterized as associated with anxiety-related personality traits [23], and according to the Harmonizome 3.0 database, *Slc18a1* is associated with the functioning of the sympathetic nervous system [24]. Considering the ISIAH rats are a model of a stress-sensitive arterial hypertension with genetically pre-determined activation of the hypothalamic–pituitary–adrenal and sympathetic adrenal systems [11], it can be assumed that the *Slc18a1* gene is a potentially interesting target for further study of its role in the development of hypertension in ISIAH rats.

Our study also identified two genes (*Cacna1s* and *Htr3a*) that determine inter-strain differences in the functioning of the postsynaptic membrane of the serotonergic synapse in the hypothalamus of hypertensive ISIAH and normotensive WAG rats at rest. *Cacna1s* encodes calcium voltage-gated channel subunit alpha1 S, which enables high-voltage-gated calcium channel activity. The members of the *CACNA* gene family may contribute to neurological phenotypes [25], but the role of *Cacna1s* in these processes has not been studied. The *Htr3a* gene encodes one of five 5-HT3 receptor subunits. 5-HT3A subunits are able to form functional homo-oligomeric receptors, which are pentameric complexes. The 5-HT3 receptor is known as a ligand-gated ion channel permeable to Na^+^, K^+^, and Ca^2+^ ions [26]. HTR3A was supposed to be associated with functional properties of brain structures central to emotion processing, particularly when exposed to stress [27]. *Htr3a* deletion prevents the occurrence of stress-induced deleterious effects [28].

According to the obtained results, *Slc18a1*, *Cacna1s*, and *Htr3a* genes can be considered as key genes modifying serotonergic synaptic transmission in the hypothalamus of hypertensive ISIAH rats. Downregulation of *Slc18a1, Cacna1s,* and *Htr3a* gene expression may be aimed at suppressing the manifestation of anxiety-related traits in ISIAH rats.

Serotonin is involved in the stress response and plays a significant role in the regulation of the neuroendocrine system and stress-related conditions. Serotonin and 5-HT agonists stimulate the release of corticotropin-releasing hormone (CRH) from hypothalamic neurons. It was described that fluoxetine promotes an increase in the CRH content in hypophyseal portal plasma and ACTH content in peripheral systemic circulation (reviewed in [19]).

ISIAH rats represent a model of stress-sensitive form of arterial hypertension and exhibit increased stress reactivity under short-term restraint stress [11]. In the present study, we examined changes in the expression of genes active in the hypothalamic serotonergic synapses of hypertensive ISIAH and normotensive WAG rats in response to a 2 h restriction of their mobility. The study revealed a group of sDEGs that unidirectionally change their expression under stress in both rat strains. Three genes (*Htr1a, Cacna1b,* and *Maoa*) were identified in the presynaptic terminal and membrane, the transcription level of which significantly decreases under stress. According to the Harmonizome 3.0 database, these three sDEGs are associated with the functioning of the sympathetic nervous system and deserve further discussion.

The 5HT1A receptor is one of the most well-studied serotonin receptors, playing a key role in serotonergic signaling. It is considered the major inhibitory serotonergic receptor associated with anxiety and depression [29]. The 5-HT1A receptor can be expressed in neurons as a presynaptic somatodendritic autoreceptor and also as a postsynaptic heteroreceptor [30]. The *Htr1a* gene expression in the hypothalamus of both rat strains decreases under stress, which is in accordance with the concept of negative regulation of neuronal serotonin (5-HT1A) receptor levels by glucocorticoids [31]. The data obtained suggest the possibility of similar functioning of 5HT1A receptors as an autoreceptor on the presynaptic membrane and as a postsynaptic heteroreceptor in the hypothalamus of both rat strains. Inter-strain differences in the stress response were also found for other serotonin receptors. In the hypothalamus of ISIAH rats, downregulation of the *Htr1b* gene is observed under stress, while in the hypothalamus of WAG rats, downregulation was found for the *Htr2a* gene.

In addition to changes in *Htr1a* gene expression, common mechanisms in the stress response include decreased expression of *Cacna1b* and *Maoa* genes in the presynaptic terminal and *Cacna1b* and *Cacna1d* genes on the postsynaptic membrane. This result suggests changes in voltage-gated calcium channel activity and a decrease in depolarization-induced calcium entry through the synaptic plasma membranes in both hypertensive and normotensive rats.

Maoa catalyzes the oxidative deamination of amines, such as dopamine, norepinephrine, and serotonin. The monoamine oxidase A has been shown to be involved in the neural response to psychosocial stress in the human brain [32]. It has been shown that genetically determined reduction in MAOA, leading to decreased clearance of catecholamines, is associated with increased response of the HPA axis to stress conditions [33,34]. Our results are in good agreement with the idea that reduced *Maoa* expression is associated with the cortisol stress response, since under the stress used in our study, the corticosterone level significantly increases in the blood plasma of both rat strains [13].

It should be noted that stress affects not only the molecular events in neuronal cells but can also affect the functioning of glial cells. Downregulation of *Gnaq*, *Plcb1,* and *Plcb2* genes is observed in the hypothalamus of both ISIAH and WAG rats. Gnaq enables G protein activity, including the G protein-coupled receptor signaling pathway. Proteins encoded by the *Plcb1* and *Plcb2* genes are activated by G proteins and catalyze the hydrolysis of phosphatidylinositol 4,5-bisphosphate to the second messengers inositol 1,4,5-trisphosphate (IP3) and diacylglycerol [35].

Considering that with the single short-term restraint stress used, plasma corticosterone increases in both rat strains, and arterial pressure increases reliably only in ISIAH rats [13], the hypothalamic genes that provide strain-specific responses to stress in hypertensive and normotensive rats are of particular interest. Genes that showed ISIAH-specific changes in expression under stress may be associated with the hypertensive status of ISIAH rats, as well as with increased stress reactivity, which is a characteristic feature of this rat strain.

The enrichment analysis of serotonergic DEG promoter regions revealed transcription factors that may be involved in the control of the common and strain-specific responses to stress. The transcription factor Tfap2a was identified as a key factor for the strain-specific stress response in the hypothalamus of the normotensive WAG rats. The key role of TFAP2A in regulating the expression of genes sensitive to 5-HT was previously identified in the scholarly work of A. Nagari, who studied the transcriptional regulatory mechanisms of 5-HT-responsive genes in embryonic stem cells [36]. The results of our study showed, for the first time, the possible involvement of Tfap2a in the regulation of the serotonergic synapse in response to a single short-term restraint stress in the hypothalamus of normotensive rats.

The key transcription factors that may be involved in the regulation of the ISIAH strain-specific hypothalamic stress response associated with the serotonergic synapse are Egr1, Rarb, and Rest.

In the central nervous system, the zinc finger transcription factor early growth response 1 (Egr1) acts as a major regulator of synaptic plasticity and neuronal activity in both physiological and pathological conditions [37]. The involvement of Egr1 in the modulation of the serotonergic system has long been known [38]. Egr1 plays a pivotal role in the response to oxidative stress and in the pathophysiology of various diseases, including atherosclerosis and hypertension [39]. In the Rat Genome Database, Egr1 is on the list of genes associated with hypertension. We have previously shown that the response to oxidative stress in the hypothalamus of ISIAH rats is more pronounced than in normotensive WAG rats [14]. The results presented in this manuscript suggest a significant role for Egr1 in the regulation of serotonergic synapse functioning in the hypothalamus of hypertensive ISIAH rats.

Rarb is a retinoic acid receptor beta, a member of the thyroid-steroid hormone receptor superfamily of nuclear transcriptional regulators. RARB levels were shown to be lower in the platelet protein profile in patients with major depression than in healthy controls [40]. It has been previously shown that retinoic acid regulates the activity of the HPA axis [41], and retinoic acid receptors might contribute to regulating the activity of CRH neurons in vivo [42]. These processes were also associated with depression-related behaviors of rats in the forced swimming test [43]. Although the association of Rarb with the regulation of the serotonergic synapse in the hypothalamus has not been reported to date, our results suggest such a possibility. In addition, since all-trans retinoic acid-induced CRH over-expression is also accompanied by arginine-vasopressin upregulation [43], the results obtained in the present study allow us to suggest that Rarb may be associated with genetically determined features of the hypertensive status in ISIAH rats.

Rest (RE1 silencing transcription factor) is known as an important factor in the regulation of monoaminergic neurotransmission, including serotonin [44]. Rest is not currently associated with hypertension, but is known to be involved in the regulation of ischemic brain injury [45] and hypoxia [46]. Our results allow us to hypothesize that Rest may be involved in the regulation of serotonergic transmission in response to restraint stress in the hypothalamus of hypertensive ISIAH rats.

In the present study, we interpreted the obtained results within the framework of the analysis of only the serotonergic synapse; however, many of the genes considered in the manuscript may participate in the regulation of the functioning of cholinergic, GABAergic, glutamatergic, and dopaminergic synapses. Accordingly, the results presented by us indicate the presence of inter-strain differences and the involvement of other synapses in the stress response as well. However, a detailed description of the molecular events at all synapses is not possible within the framework of a single manuscript.

Another limitation of this study is that it was conducted only on males. Further research is needed to understand whether the results presented in the manuscript could be applied to females.

In general, the following may be concluded:

Expression of genes in serotoninergic synapses of the hypothalamus of hypertensive ISIAH and normotensive WAG rats may demonstrate a high level of inter-strain differences. The authors suggest that the differences found in the transcriptional activity of some genes associated with serotonergic regulation of neuroendocrine and stress reactions may relate to the function of the cardiovascular system and the regulation of blood pressure.

## 4. Materials and Methods

### 4.1. Animals

The experiment was carried out on three-month-old male hypertensive rats of the ISIAH/Icgn (inherited stress-induced arterial hypertension) strain and normotensive rats of the WAG/GSto-Icgn (Wistar Albino Glaxo) strain. Rats were kept under standard conditions on a 12-h light-dark schedule (LD 12:12) in a conventional vivarium of the Center for Genetic Resources of Laboratory Animals of the Institute of Cytology and Genetics, Siberian Branch of the Russian Academy of Sciences. There were no restrictions on water and balanced food. The experimental procedures, including animal handling, stress protocol, tissue collection, and RNA isolation, were identical for all rats and are described in detail in our earlier publication [13]. In the current work, we conducted a secondary, pathway-specific analysis focusing on genes related to serotonergic signaling.

For hypothalamic transcriptome analysis by RNA-Seq, four groups of seven animals each were formed: (1) ISIAH _control; (2) WAG_control; (3) ISIAH _stress; (4) WAG_stress. In all rats, basal systolic blood pressure (BP) was measured by the tail-cuff method [11] under light ether anesthesia to prevent emotional stress during the measurement. Seven days after basal BP measurement, a 2 h restraint stress was performed for rats of two experimental groups, ISIAH_stress and WAG_stress. The stressing procedure consisted of placing the rat in a tight-wire-mesh cage for 2 h. Thirty minutes before the end of the stress procedure, the cage with the rat was placed on a warm (37 °C) platform to prepare the rat for blood pressure measurement. Immediately after the end of the stress procedure, rats were measured for BP (without anesthesia) and immediately decapitated, hypothalamus was isolated and homogenized in 700 μL of ExtractRNA reagent (Evrogen, Moscow, Russia) using 500 μL of Lysing matrix D (Cat#6540434 MP Biomedicals, Solon, OH, USA) for 20 s at 18,000 rpm in a Super FastPrep-2 homogenizer (MP Biomedicals, Solon, OH, USA). All procedures were conducted in compliance with European Communities Council Directive 210/63/EU of 22 September 2010. The study protocol was approved by the Bioethical Council of the federal research center, Institute of Cytology and Genetics SB RAS (Novosibirsk, Russia), protocol No. 115 of 20 December 2021.

### 4.2. RNA-Seq

Sample preparation and sequencing were performed as described previously [13]. RNA isolation was performed at the Institute of Genomic Analysis (Moscow, Russia). Sample preparation and sequencing of hypothalamic transcriptomes according to the manufacturer’s protocols (MGI Tech Co., Ltd., Shenzhen, China) were performed at BGI Hong Kong Tech Solution NGS Lab (Hong Kong, China). Pair-end sequencing of cDNA libraries was performed on the DNBSEQ platform (DNBSEQ Technology, Hong Kong, China) with a read length of 150 base pairs and sequencing depth of more than 30 million uniquely mapped reads. All samples were analyzed as biological replicates.

The quality of the obtained sequencing data was assessed using the FastQC program (version 0.11.5 [47]). The total number of nucleotide reads for the libraries after filtering was 1,287,393,367, of which 1,267,436,623 nucleotide reads (98.45%) were mapped to the reference rat genome mRatBN7.2/rn7 (rn7 assembly Wellcome Sanger Institute Nov, 2020) using the STAR software package, version 2.7.10a [48].

Statistical analysis to calculate differential gene expression was performed in the R statistical computing environment. We applied surrogate variable analysis (SVA) [49] to account for unwanted variation in the data caused by possible random systematic bias in the sample preparation process. For SVA analysis, expression data were normalized and transformed using the vst function in DESeq2 v1.30.1 [50] according to the documentation. Significant surrogate variables were further included as factors in the differential expression analysis in DESeq2. Differential expression analysis was performed between the comparison groups ISIAH_stress-ISIAH_rest and WAG_stress-WAG_rest. Hypothalamic samples from seven rats were analyzed in each of the four groups.

Differential expression was calculated for all genes showing sufficient expression levels above a threshold (sum of gene coverage for all libraries greater than 10 reads). The significance threshold for identifying differentially expressed genes was chosen with correction for multiple comparisons and corresponded to an adjusted *p*-value < 5%. The comparative transcription level of the identified serotonergic DEGs is presented in the Appendix A. Validation of the sequencing results (Appendix A) was performed using inter-strain comparison data at rest from previously performed RNA-seq of the hypothalamus of 3-month-old male ISIAH and WAG rats [51].

### 4.3. Functional Annotation of DEGs

Functional annotation of DEGs was carried out using the KEGG Pathway Database [52], Rat Genome Database [53], DAVID (The Database for Annotation, Visualization and Integrated Discovery) [54], STRING database [55], and Harmonizome 3.0 database [24]. The enrichment analysis of gene promoter regions with transcription factor binding sites was performed using the Enrichr resource [56].

### 4.4. Visualization and Graphing

The SRplot, a free online platform, was used for data visualization and graphing [57]. Cytoscape Web was used to construct the biomolecular interaction networks [58].

## Figures and Tables

**Figure 1 ijms-26-07058-f001:**
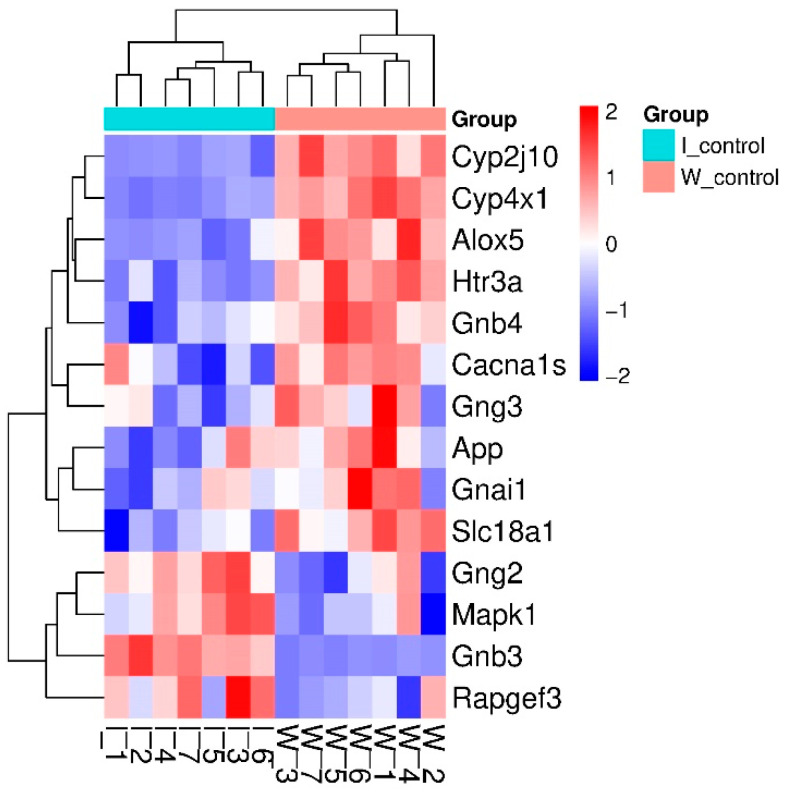
Clustering of sDEGs representing inter-strain differences in resting state. Most of the sDEGs have decreased transcription levels in the hypothalamus of hypertensive ISIAH rats.

**Figure 2 ijms-26-07058-f002:**
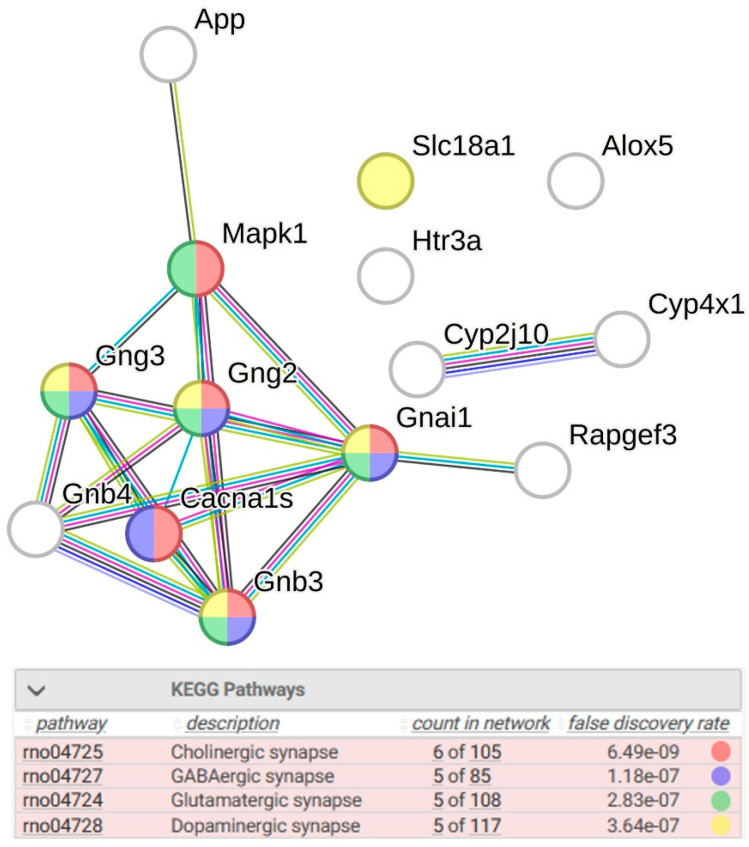
DEGs associated with the serotonergic synapse, which may also be involved in the control of cholinergic, GABAergic, glutamatergic, and dopaminergic synapse function. PPI enrichment *p*-value: 4 × 10^−15^. Purple lines indicate experimentally determined interactions; blue lines denote known interactions from curated databases; dark blue lines represent gene co-occurrence; black lines indicate co-expression; green lines represent the results of text mining; and gray lines indicate the presence of protein homology.

**Figure 3 ijms-26-07058-f003:**
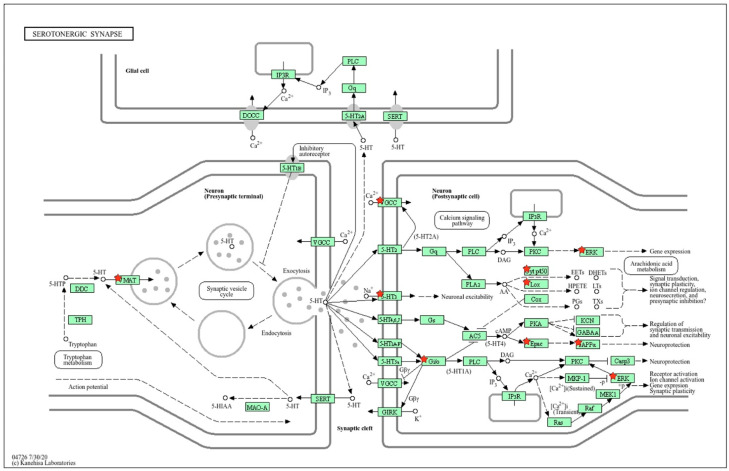
Proteins encoded by sDEGs determining inter-strain differences at rest. The key sDEGs determining inter-strain differences in resting state are the *Slc18a1* gene encoding the vesicular monoamine transporter (VMAT) in the presynaptic terminal and the *Cacna1s* and *Htr3a* genes at the postsynaptic membrane. *Cacna1s* enables voltage-gated calcium channel (VGCC) activity. The 5-HT3 (*Htr3a* gene) belongs to the ligand-gated ion channel receptor superfamily. These differences may determine a cascade of further changes in postsynaptic signaling pathways at the hypothalamic serotonergic synapse of hypertensive ISIAH and normotensive WAG rats. Proteins encoded by sDEGs are shown as red stars on the KEGG map for the serotonergic synapse. The schematic image was built in the KEGG database.

**Figure 4 ijms-26-07058-f004:**
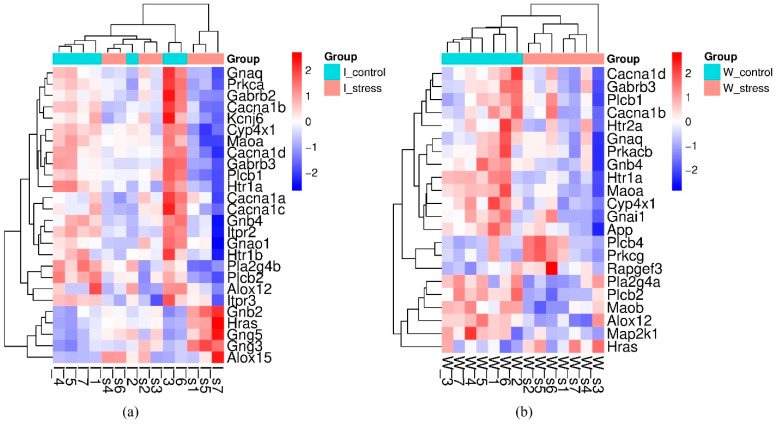
Clustering of DEGs associated with stress response in the serotonergic synapse of the hypothalamus of (**a**) hypertensive ISIAH rats; (**b**) normotensive WAG rats. Most of the sDEGs have decreased transcription levels in the hypothalamus of both rat strains when exposed to a single short-term (2 h) restraint stress.

**Figure 5 ijms-26-07058-f005:**
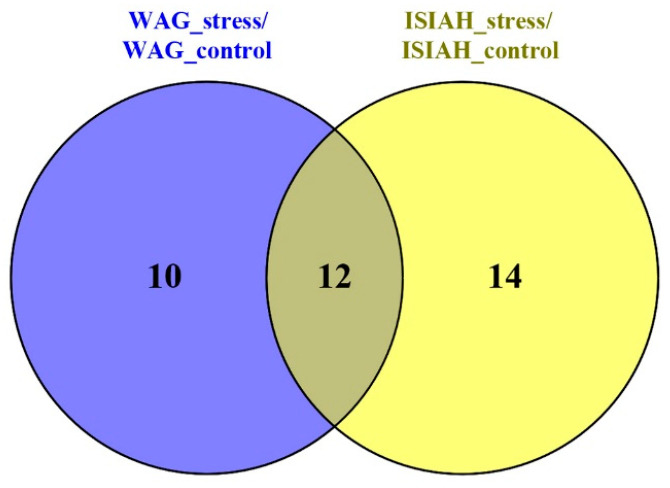
Comparison of the stress response in the serotonergic synapse of the hypothalamus in rats of two strains.

**Figure 6 ijms-26-07058-f006:**
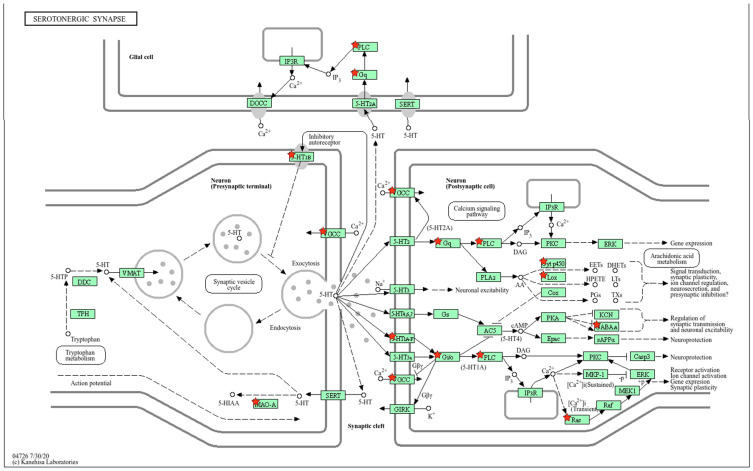
Common response to stress in the serotonergic synapse of the hypothalamus of hypertensive ISIAH and normotensive WAG rats is associated with downregulation of the *Htr1a*, *Cacna1b*, and *Maoa* genes in the presynaptic region (the corresponding proteins in the figure are indicated as 5-HT1B, VGCC, and MAO-A). Changes in the functioning of the postsynaptic membrane are associated with a decrease in the expression of the *Cacna1b*, *Cacna1d*, and *Htr1a* genes (the corresponding proteins in the figure are indicated as 5-HT1A-F and VGCC). A further cascade of molecular events in postsynaptic signaling of the serotonergic synapse and in glial cells occurs through downregulation of the *Gnaq*, *Plcb1*, and *Plcb2* genes (the corresponding proteins in the figure are indicated as Gq and PLC). Proteins encoded by sDEGs are shown as red stars. The schematic image was built in the KEGG database.

**Figure 7 ijms-26-07058-f007:**
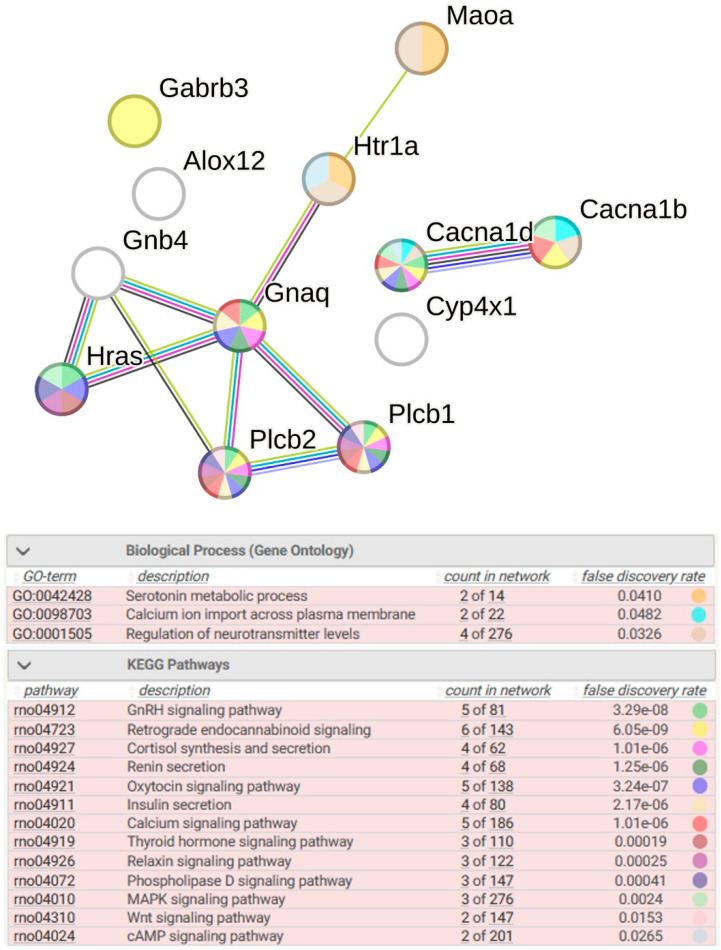
Functional enrichments in the network of sDEGs related to a common response to stress. PPI enrichment *p*-value: 4.32 × 10^−7^. Purple lines indicate experimentally determined interactions; blue lines denote known interactions from curated databases; dark blue lines represent gene co-occurrence; black lines indicate co-expression; green lines represent the results of text mining; and gray lines indicate the presence of protein homology.

**Figure 8 ijms-26-07058-f008:**
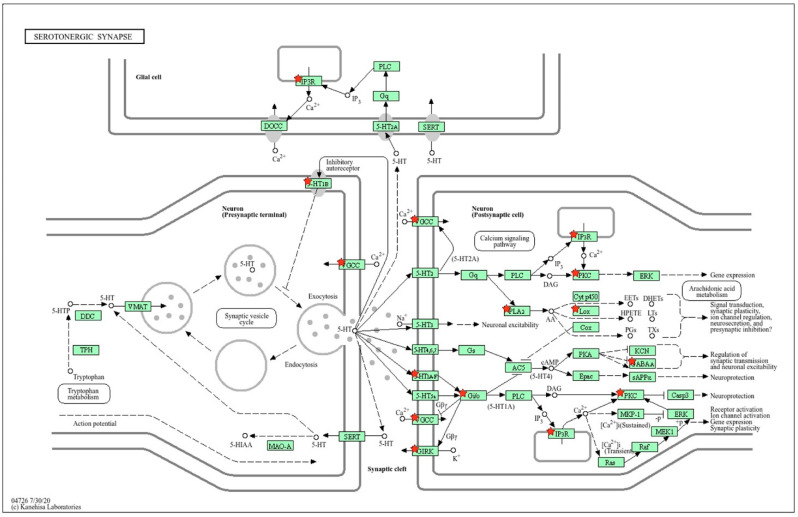
Proteins encoded by ISIAH-specific sDEGs (red stars) identified on the KEGG map for serotonergic synapse. Strain-specific response to stress in the serotonergic synapse of the hypothalamus of hypertensive ISIAH rats is associated with downregulation of the *Cacna1a* and *Htr1b* genes on the presynaptic membrane (the corresponding proteins in the figure are indicated as VGCC and 5-HT1B). Changes in the functioning of the postsynaptic membrane are associated with a decrease in the expression of the *Cacna1a*, *Cacna1c*, *Htr1b*, and *Kcnj6* genes (the corresponding proteins in the figure are indicated as VGCC, 5-HT1A-F, and GIRK). In the subsequent cascade of molecular events in serotonergic postsynaptic signaling, as well as in glial cells, downregulation of the *Itpr2* and *Itpr3* genes plays a key role (the corresponding protein in the figure is indicated as IP3R). The schematic image was built in the KEGG database.

**Figure 9 ijms-26-07058-f009:**
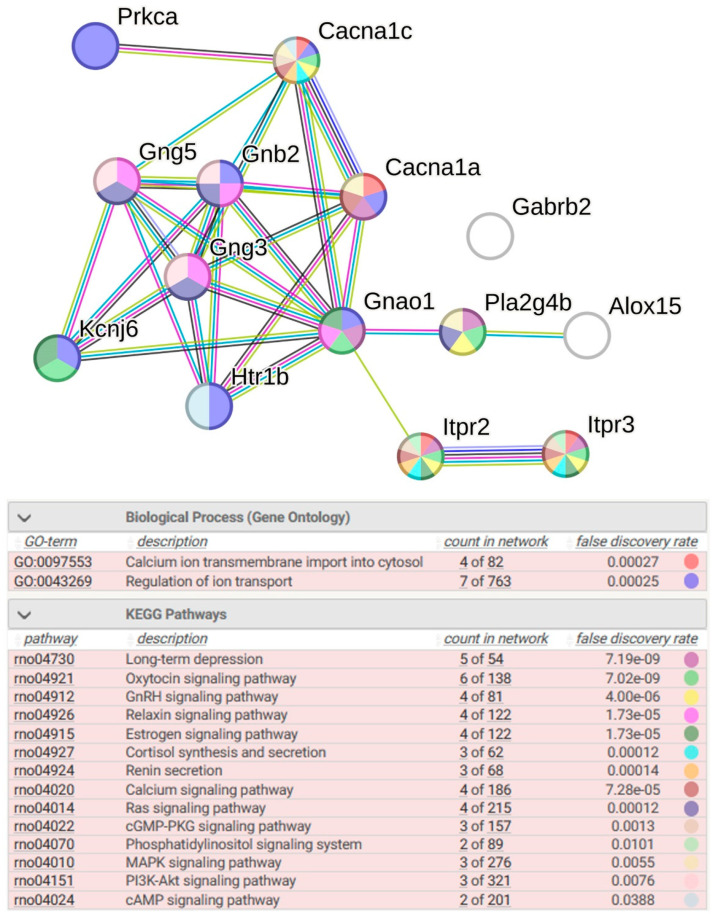
Functional enrichments in the network of sDEGs related to ISIAH-specific response to stress. PPI enrichment *p*-value: < 1.0 × 10^−16^. Purple lines indicate experimentally determined interactions; blue lines denote known interactions from curated databases; dark blue lines represent gene co-occurrence; black lines indicate co-expression; green lines represent the results of text mining; and gray lines indicate the presence of protein homology.

**Figure 10 ijms-26-07058-f010:**
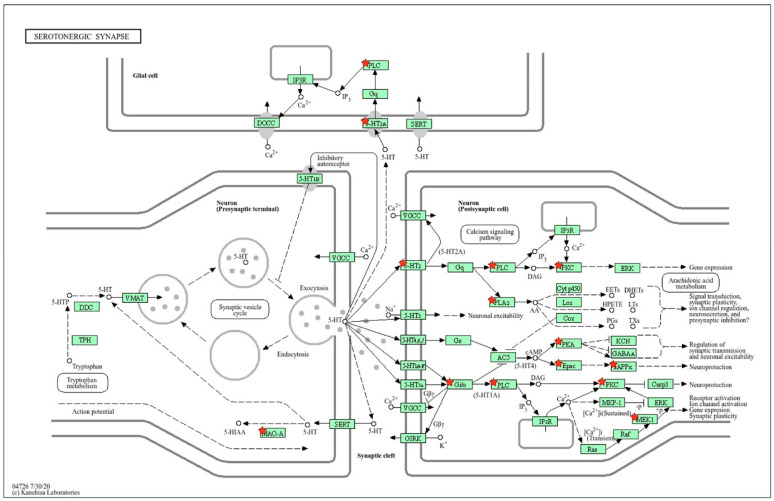
Proteins encoded by WAG-specific sDEGs (red stars) are identified on the KEGG map for the serotonergic synapse. Strain-specific response to stress in the hypothalamic serotonergic synapse of normotensive WAG rats is associated with downregulation of the *Maob* gene in the presynaptic region (the corresponding protein in the figure is indicated as MAO-A). Changes in the functioning of the postsynaptic membrane, as well as glial cells, are associated with a decrease in the expression of the *Htr2a* gene, which is related to an increase in *Plcb4*, catalyzing the formation of inositol 1,4,5-trisphosphate and diacylglycerol from phosphatidylinositol 4,5-bisphosphate (the corresponding proteins in the figure are indicated as 5-HT2 and PLC). The schematic image was built in the KEGG database.

**Figure 11 ijms-26-07058-f011:**
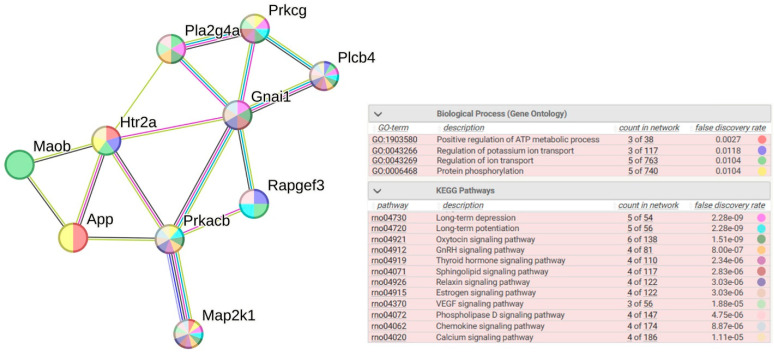
Functional enrichments in the network of sDEGs related to WAG-specific response to stress. PPI enrichment *p*-value: 8.69 × 10^−8^. Purple lines indicate experimentally determined interactions; blue lines denote known interactions from curated databases; dark blue lines represent gene co-occurrence; black lines indicate co-expression; green lines represent the results of text mining; and gray lines indicate the presence of protein homology.

**Figure 12 ijms-26-07058-f012:**
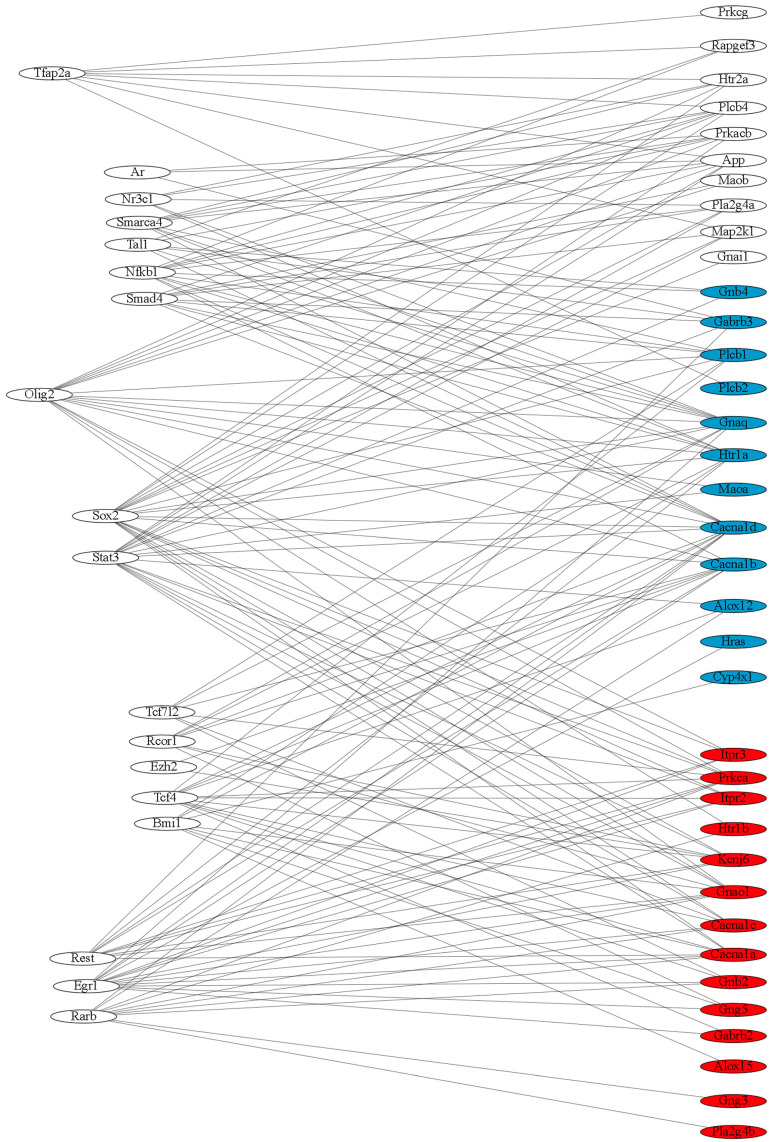
The biomolecular interaction networks between transcription factors and serotonergic DEGs. On the left side of the figure are the transcription factors, and on the right side of the figure are the genes that have binding sites for the corresponding transcription factors in the promoter region. The serotonergic DEGs representing the common stress response group in the hypothalamus of ISIAH and WAG rats are shown in blue. The serotonergic DEGs representing the WAG-specific stress response are shown in white; the serotonergic DEGs representing the ISIAH-specific stress response are shown in red.

**Table 1 ijms-26-07058-t001:** DEGs associated with the function of the serotonergic synapse in the hypothalamus of hypertensive ISIAH rats and normotensive WAG rats at rest.

Gene Symbol	GeneID	log2 Fold Change ISIAH_Control/ WAG_Control	padj	Description
*Alox5 **	25290	−0.641	5.14 × 10^−6^	arachidonate 5-lipoxygenase
*App **	54226	−0.138	3.06 × 10^−6^	amyloid beta precursor protein
*Cacna1s*	682930	−1.110	4.83 × 10^−2^	calcium voltage-gated channel subunit alpha1 S
*Cyp2j10*	313373	−1.158	1.30 × 10^−21^	cytochrome P450, family 2, subfamily j, polypeptide 10
*Cyp4x1*	246767	−0.982	1.55 × 10^−52^	cytochrome P450, family 4, subfamily x, polypeptide 1
*Gnai1*	25686	−0.108	2.07 × 10^−2^	G protein subunit alpha i1
*Gnb3 *#*	60449	2.397	3.27 × 10^−49^	G protein subunit beta 3
*Gnb4*	294962	−0.369	7.44 × 10^−6^	G protein subunit beta 4
*Gng2*	80850	0.111	1.66 × 10^−2^	G protein subunit gamma 2
*Gng3*	114117	−0.191	7.59 × 10^−4^	G protein subunit gamma 3
*Htr3a #*	79246	−0.509	1.00 × 10^−8^	5-hydroxytryptamine receptor 3A
*Mapk1 **	116590	0.137	2.12 × 10^−3^	mitogen-activated protein kinase 1
*Rapgef3*	59326	0.121	2.61 × 10^−3^	Rap guanine nucleotide exchange factor 3
*Slc18a1 #*	25693	−0.638	1.57 × 10^−2^	solute carrier family 18 member A1

* DEGs associated with hypertension (according to Rat Genome Database); #—genes for sympathetic nervous system (according to Harmonizome 3.0).

**Table 2 ijms-26-07058-t002:** Common response to stress.

Gene Symbol	GeneID	ISIAH_Stress/ISIAH_Control		WAG_Stress/WAG_Control		Description
		log2 Fold Change	padj	log2 Fold Change	padj	
*Alox12 **	287454	−0.716	3.07 × 10^−2^	−0.779	8.06 × 10^−3^	arachidonate 12-lipoxygenase, 12S type
*Cacna1b #*	257648	−0.290	8.82 × 10^−9^	−0.152	1.28 × 10^−4^	calcium voltage-gated channel subunit alpha1 B
*Cacna1d*	29716	−0.300	2.82 × 10^−4^	−0.287	2.02 × 10^−6^	calcium voltage-gated channel subunit alpha1 D
*Cyp4x1*	246767	−0.363	1.15 × 10^−3^	−0.261	6.10 × 10^−5^	cytochrome P450, family 4, subfamily x, polypeptide 1
*Gabrb3*	24922	−0.652	2.88 × 10^−5^	−0.405	2.19 × 10^−3^	gamma-aminobutyric acid type A receptor subunit beta 3
*Gnaq #*	81666	−0.480	1.41 × 10^−3^	−0.307	1.59 × 10^−2^	G protein subunit alpha q
*Gnb4*	294962	−0.320	1.19 × 10^−2^	−0.272	9.44 × 10^−3^	G protein subunit beta 4
*Hras*	293621	0.227	5.62 × 10^−4^	0.106	4.06 × 10^−2^	HRas proto-oncogene, GTPase
*Htr1a #*	24473	−0.575	3.95 × 10^−3^	−0.639	9.84 × 10^−6^	5-hydroxytryptamine receptor 1A
*Maoa #*	29253	−0.162	9.73 × 10^−3^	−0.230	2.83 × 10^−6^	monoamine oxidase A
*Plcb1*	24654	−0.286	7.57 × 10^−3^	−0.154	4.39 × 10^−2^	phospholipase C beta 1
*Plcb2*	85240	−0.431	3.41 × 10^−2^	−0.596	2.20 × 10^−4^	phospholipase C, beta 2

* DEGs associated with hypertension (according to Rat Genome Database); #—genes for sympathetic nervous system (according to Harmonizome 3.0).

**Table 3 ijms-26-07058-t003:** sDEGs representing the strain-specific stress response in the hypothalamus of ISIAH rats.

Gene Symbol	GeneID	log2 Fold Change ISIAH_Stress/ ISIAH_Control	padj	Description
*Alox15 **	81639	1.236	1.98 × 10^−2^	arachidonate 15-lipoxygenase
*Cacna1a*	25398	−0.162	5.36 × 10^−3^	calcium voltage-gated channel subunit alpha1 A
*Cacna1c **	24239	−0.211	3.96 × 10^−2^	calcium voltage-gated channel subunit alpha1 C
*Gabrb2*	25451	−0.621	1.19 × 10^−2^	gamma-aminobutyric acid type A receptor subunit beta 2
*Gnao1*	50664	−0.232	1.66 × 10^−3^	G protein subunit alpha o1
*Gnb2*	81667	0.162	1.62 × 10^−6^	G protein subunit beta 2
*Gng3*	114117	0.185	4.89 × 10^−3^	G protein subunit gamma 3
*Gng5*	79218	0.150	3.08 × 10^−2^	G protein subunit gamma 5
*Htr1b **	25075	−0.451	3.33 × 10^−2^	5-hydroxytryptamine receptor 1B
*Itpr2 **	81678	−0.271	2.16 × 10^−3^	inositol 1,4,5-trisphosphate receptor, type 2
*Itpr3 *#*	25679	−0.389	9.87 × 10^−4^	inositol 1,4,5-trisphosphate receptor, type 3
*Kcnj6*	25743	−0.168	7.57 × 10^−3^	potassium inwardly-rectifying channel, subfamily J, member 6
*Pla2g4b*	311341	−0.334	1.01 × 10^−3^	phospholipase A2 group IVB
*Prkca **	24680	−0.357	1.33 × 10^−2^	protein kinase C, alpha

* DEGs associated with hypertension (according to Rat Genome Database); #—genes for sympathetic nervous system (according to Harmonizome 3.0).

**Table 4 ijms-26-07058-t004:** sDEGs representing the strain-specific stress response in the hypothalamus of WAG rats.

Gene Symbol	GeneID	log2 Fold Change WAG_Stress/ WAG_Control	padj	Description
*App **	54226	−0.197	1.13 × 10^−8^	amyloid beta precursor protein
*Gnai1*	25686	−0.147	5.01 × 10^−3^	G protein subunit alpha i1
*Htr2a **	29595	−0.388	4.97 × 10^−2^	5-hydroxytryptamine receptor 2A
*Maob #*	25750	−0.224	1.04 × 10^−5^	monoamine oxidase B
*Map2k1*	170851	−0.106	1.50 × 10^−2^	mitogen-activated protein kinase kinase 1
*Pla2g4a*	24653	−0.294	1.85 × 10^−2^	phospholipase A2 group IVA
*Plcb4*	25031	0.337	2.73 × 10^−2^	phospholipase C, beta 4
*Prkacb*	293508	−0.284	1.04 × 10^−3^	protein kinase cAMP-activated catalytic subunit beta
*Prkcg*	24681	0.343	3.60 × 10^−2^	protein kinase C, gamma
*Rapgef3*	59326	0.083	3.34 × 10^−2^	Rap guanine nucleotide exchange factor 3

* DEGs associated with hypertension (according to Rat Genome Database); #—genes for sympathetic nervous system (according to Harmonizome 3.0).

## Data Availability

The RNA-Seq data analyzed in the present study were obtained in a previously described experiment [13], which investigated hypothalamic transcriptomic responses to restraint stress in ISIAH and WAG rats. The results obtained in the study were deposited in the RatDEGdb database [59] and are available at https://www.sysbio.ru/RatDEGdb (accessed on 10 January 2025). In the current work, we performed a secondary, pathway-specific analysis focusing on genes related to serotonergic signaling. All data related to this publication are available in the manuscript and Appendix A.

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
