# Peer review of "Expression Profiles of Genes Related to Serotonergic Synaptic Function in Hypothalamus of Hypertensive and Normotensive Rats in Basal and Stressful Conditions"

_ijms, 2025, doi:10.3390/ijms26157058_

Round 1

Reviewer 1 Report

Comments and Suggestions for Authors

This manuscript,”Expression Profiles of Genes Related to Serotonergic Synaptic 2 Function in Hypothalamus of Hypertensive and Normotensive 3 Rats at the Basal and Stressful Conditions” presents a focused transcriptomic analysis of serotonergic signaling pathways in the hypothalamus of hypertensive ISIAH and normotensive WAG rats, following acute emotional stress. The authors build on previously generated RNA-Seq data, narrowing the scope of interpretation to serotonergic synapse-related gene expression. This targeted approach allows for deeper insights into the neurochemical mechanisms underlying strain-specific stress responses and offers potentially valuable implications for understanding the neuroendocrine regulation of blood pressure. The manuscript provides a valuable contribution by highlighting serotonergic components of the stress response that may differentiate hypertensive and normotensive phenotypes.

However, it is essential to emphasize that the presented data originate from the same experiment and animal material as previously published work: ”Effect of Short-Term Restraint Stress on the Hypothalamic Transcriptome Profiles of Rats with Inherited Stress-Induced Arterial Hypertension (ISIAH) and Normotensive Wistar Albino Glaxo (WAG) Rats” (Oshchepkov  et al. 2024, reference 32). Therefore, to meet transparency and good publication practice standards, the manuscript must clearly acknowledge this in multiple sections. Specifically:

  1. The Introduction, Methods, and Discussion must include explicit statements clarifying that this is a secondary, pathway-focused analysis based on previously published transcriptomic data.

Here is a paragraph that could be included as a preamble to the Materials and Methods section (as a suggestion only, to be used or adapted at the authors’ discretion):

”The RNA-Seq data analyzed in the present study were obtained in a previously described experiment (Oshchepkov  et al. 2024), which investigated hypothalamic transcriptomic responses to restraint stress in ISIAH and WAG rats. In the current work, we performed a secondary, pathway-specific analysis focusing on genes related to serotonergic signaling. The experimental procedures, including animal handling, stress protocol, tissue collection, and RNA isolation, were identical to those detailed in our earlier publication (Oshchepkov  et al. 2024).”

  1. The Funding statement must be corrected. The current declaration—"This research received no external funding in 2025"—is inappropriate and misleading, given that the original experiment was supported by research funding, as stated in the Acknowledgments. The correct source of funding for the generation of the omics data must be transparently disclosed.
  2. In addition, the Data Availability Statement should include access information or references to where the raw data were previously published or deposited, to ensure traceability.

Only after these formal requirements are fully addressed can the manuscript be considered for publication.

Author Response

The authors are grateful to the reviewer for carefully reading the manuscript and recommendations for its improvement. All corrections made to the text of the manuscript are shown in red.

Comments and Suggestions for Authors

This manuscript,”Expression Profiles of Genes Related to Serotonergic Synaptic  Function in Hypothalamus of Hypertensive and Normotensive  Rats at the Basal and Stressful Conditions” presents a focused transcriptomic analysis of serotonergic signaling pathways in the hypothalamus of hypertensive ISIAH and normotensive WAG rats, following acute emotional stress. The authors build on previously generated RNA-Seq data, narrowing the scope of interpretation to serotonergic synapse-related gene expression. This targeted approach allows for deeper insights into the neurochemical mechanisms underlying strain-specific stress responses and offers potentially valuable implications for understanding the neuroendocrine regulation of blood pressure. The manuscript provides a valuable contribution by highlighting serotonergic components of the stress response that may differentiate hypertensive and normotensive phenotypes.

However, it is essential to emphasize that the presented data originate from the same experiment and animal material as previously published work: ”Effect of Short-Term Restraint Stress on the Hypothalamic Transcriptome Profiles of Rats with Inherited Stress-Induced Arterial Hypertension (ISIAH) and Normotensive Wistar Albino Glaxo (WAG) Rats” (Oshchepkov  et al. 2024, reference 32). Therefore, to meet transparency and good publication practice standards, the manuscript must clearly acknowledge this in multiple sections. Specifically:

  1. The Introduction, Methods, and Discussion must include explicit statements clarifying that this is a secondary, pathway-focused analysis based on previously published transcriptomic data.

Here is a paragraph that could be included as a preamble to the Materials and Methods section (as a suggestion only, to be used or adapted at the authors’ discretion):

”The RNA-Seq data analyzed in the present study were obtained in a previously described experiment (Oshchepkov  et al. 2024), which investigated hypothalamic transcriptomic responses to restraint stress in ISIAH and WAG rats. In the current work, we performed a secondary, pathway-specific analysis focusing on genes related to serotonergic signaling. The experimental procedures, including animal handling, stress protocol, tissue collection, and RNA isolation, were identical to those detailed in our earlier publication (Oshchepkov  et al. 2024).”

Answer: The statements clarifying that this is a secondary, pathway-focused analysis based on previously published transcriptomic data were included in the Introduction (lines 97-103), Methods (lines 531-532), and Discussion (line 364). Additionally, this is also explained in the Data Availability Statement (lines 612-618).

  1. The Funding statement must be corrected. The current declaration—"This research received no external funding in 2025"—is inappropriate and misleading, given that the original experiment was supported by research funding, as stated in the Acknowledgments. The correct source of funding for the generation of the omics data must be transparently disclosed.

Answer: Done (lines 606-607).

  1. In addition, the Data Availability Statement should include access information or references to where the raw data were previously published or deposited, to ensure traceability.

Answer: The authors have supplemented the information in the Data Availability Statement section (Lines 612-618).

Only after these formal requirements are fully addressed can the manuscript be considered for publication.

Reviewer 2 Report

Comments and Suggestions for Authors

This paper is a transcriptomic analysis of the genes related to serotonergic synapse in the hypothalamus of hypertensive ISIAH rats and normotensive WAG rats in both basal and acute restraint stress conditions. The study is comprehensive, and the analysis pipeline is stringent, integrating KEGG pathway mapping, functional annotation, and differential gene expression analysis. However, some fundamental problems of data interpretation, presentation clarity, and biological validation need to be addressed to enhance the scientific content and readability of the findings.

1. The study focuses on a less explored aspect of hypertension, central serotonergic signaling, more so in the context of stress. However, novelty can be better emphasized in the context of existing transcriptomic data in ISIAH rats.

2. The introduction and abstract adequately mention the aims. Nevertheless, it would be advantageous to provide a brief explanation for why serotonergic synapses in the hypothalamus are a focal point in particular rather than other monoaminergic systems.

3. Although n=7 per group is acceptable for RNA-Seq, we are not informed whether power analysis was performed. Can you please indicate whether the sample size is sufficient to detect biologically relevant differences?

4. There is no experimental validation (e.g., qPCR or protein level) of significant differentially expressed genes. At least 2–3 genes from each category (resting, stress, strain-specific) must be validated to make the results more solid.

5. The discussion is pathway analysis-rich but not as much in terms of direct biological interpretation. How, for example, would downregulation of Slc18a1 or Htr3a mechanistically result in hypertension or stress susceptibility?

6. The manuscript uses gene symbols inconsistently (e.g., Slc18a1, SLC18A1). Please use consistent gene nomenclature throughout in line with standard rat gene conventions.

7. The rationale for the use of a 2-hour restraint stress model needs to be clarified. How does this model mimic emotional stress in humans, and what were the reasons for its choice over chronic or other acute stress models?

8. Were handling variables, food intake, or circadian factors controlled for prior to tissue collection? They could potentially influence hypothalamic gene expression significantly.

9. While KEGG pathway maps are useful, other functional enrichment results (e.g., from GO analysis or Reactome) would expand the interpretation to more than just serotonergic synapses.

10. Figures are crowded and sometimes difficult to interpret. For instance, Figure 3 should have had pre- vs. post-synaptic compartments clearly indicated. The addition of simplified schematic summary diagrams of key pathways would be beneficial.

11. Terminology such as "strain-specific stress response" needs to be defined clearly at the beginning. Furthermore, it is convenient to explain abbreviations, such as sDEGs, upon their first use in the Results section.

12. Male rats only were used in this experiment. Note the sex limitation and whether these results could be applied to females.

13. Reference [12] appears to report overlapping transcriptomic results. Can you please describe how this manuscript reports distinct conclusions from that work?

14. The identification of TFs like Egr1 and Tfap2a is intriguing. Are there any additional details or citations to prior studies that implicated these TFs in serotonergic regulation or hypertension?

15. A few of the conclusions, especially linking expression changes to functional cardiovascular consequences, seem speculative. Would you soften or support these comments with additional data or citations?

Comments on the Quality of English Language

The manuscript is generally readable and written in clear academic English. However, there are several instances where grammar, sentence structure, or word choice could be improved for clarity and flow.

Author Response

The authors are grateful to the reviewer for carefully reading the manuscript and recommendations for its improvement. All corrections made to the text of the manuscript are shown in red.

Comments and Suggestions for Authors

This paper is a transcriptomic analysis of the genes related to serotonergic synapse in the hypothalamus of hypertensive ISIAH rats and normotensive WAG rats in both basal and acute restraint stress conditions. The study is comprehensive, and the analysis pipeline is stringent, integrating KEGG pathway mapping, functional annotation, and differential gene expression analysis. However, some fundamental problems of data interpretation, presentation clarity, and biological validation need to be addressed to enhance the scientific content and readability of the findings.

  1. The study focuses on a less explored aspect of hypertension, central serotonergic signaling, more so in the context of stress. However, novelty can be better emphasized in the context of existing transcriptomic data in ISIAH rats.

Answer: The Introduction section has been supplemented with text emphasizing the novelty of the analysis presented in the manuscript in the context of existing transcriptomic data in ISIAH rats (lines 97-103).

  1. The introduction and abstract adequately mention the aims. Nevertheless, it would be advantageous to provide a brief explanation for why serotonergic synapses in the hypothalamus are a focal point in particular rather than other monoaminergic systems.

Answer: The Introduction section has been updated to include text explaining why the manuscript discusses the serotonergic synapse rather than other monoaminergic systems (lines 103-107).

  1. Although n=7 per group is acceptable for RNA-Seq, we are not informed whether power analysis was performed. Can you please indicate whether the sample size is sufficient to detect biologically relevant differences?

Answer: We have previously performed the power analysis for this data for the original paper (Oshchepkov et al., 2024) with the tool RNASeqPower (Therneau et al., 2021, Calculating samplesSize estimates for RNA Seq studies. R package version 1.32.0.).  Having an average depth of coverage for the transcript or gene more than 64, sample size of 7 in each group and typical biological coefficient of variation for inbred animal lines 0.1 we obtained a power estimation 0.997.

  1. There is no experimental validation (e.g., qPCR or protein level) of significant differentially expressed genes. At least 2–3 genes from each category (resting, stress, strain-specific) must be validated to make the results more solid.

Answer: The authors deliberately performed the experiment on sufficiently representative groups of animals (n=7) and with good sequencing depth to be able to reliably analyze all genes, including low-expressed ones, and to avoid the need to confirm the expression of key genes by qPCR. However, the authors realize that validation of the results is always desirable in experimental work. Our research group has transcriptome sequencing data from hypothalamus of ISIAH and WAG males of the same age, with interstrain comparisons at rest. This experiment was performed earlier (10 years ago). In that study, a total of 139 DEGs were identified, two of which were associated with the function of the serotonergic synapse according to the KEGG database. Comparative analysis of the RNA-seq protocols and the obtained results of the two experiments (performed in 2014 and 2024) are presented in Supplementary Table S5. Although the RNA-seq analysis protocols in the two experiments differed greatly, the results from the two studies were in good agreement.

Transcriptome analysis was performed to screen key genes that are interesting for further analysis at the protein level and potentially promising for use in preclinical studies. The authors agree with the reviewer that analysis of key genes at the protein level can certainly be informative; however, to perform it, it is necessary to once again prepare four groups of animals according to the same protocol to be able to collect hypothalamus for protein level analysis. Unfortunately, for a number of reasons, primarily financial, the authors currently do not have the opportunity to conduct such an experiment and analyze the expression of proteins encoded by key genes using Western blot or enzyme immunoassay.

  1. The discussion is pathway analysis-rich but not as much in terms of direct biological interpretation. How, for example, would downregulation of Slc18a1 or Htr3a mechanistically result in hypertension or stress susceptibility?

Answer: The Discussion section has been corrected according to the reviewer's recommendations (lines 399-404).

  1. The manuscript uses gene symbols inconsistently (e.g., Slc18a1, SLC18A1). Please use consistent gene nomenclature throughout in line with standard rat gene conventions.

Answer: Thank you very much. The correction has been made throughout the text.

  1. The rationale for the use of a 2-hour restraint stress model needs to be clarified. How does this model mimic emotional stress in humans, and what were the reasons for its choice over chronic or other acute stress models?

Answer: The Introduction section has been updated with text explaining the similarity of neuroendocrine changes under stress in ISIAH rats with those under emotional stress in humans (lines 85-86) and the choice of stress duration in the study (Lines 87-92).

  1. Were handling variables, food intake, or circadian factors controlled for prior to tissue collection? They could potentially influence hypothalamic gene expression significantly.

Answer: All rats were kept identically. The Methods section has been updated to explain this (lines 529-531).

  1. While KEGG pathway maps are useful, other functional enrichment results (e.g., from GO analysis or Reactome) would expand the interpretation to more than just serotonergic synapses.

Answer: This article was written as a result of an invitation to a special issue of “Serotonin in Health and Diseases”. The authors plan to describe other monoaminergic systems in detail in separate publications. The Introduction section contains explanatory text (lines 103-107).

  1. Figures are crowded and sometimes difficult to interpret. For instance, Figure 3 should have had pre- vs. post-synaptic compartments clearly indicated. The addition of simplified schematic summary diagrams of key pathways would be beneficial.

Answer: The authors used standard diagrams from the KEGG database. The authors agree that in these diagrams pre- and post-synaptic compartments are indicated in very small print, but we are not sure that we can make changes to the authors' diagrams from the KEGG database. From the point of view of the authors, the diagrams from the KEGG database reflect a complete picture of the changed and unchanged pathways, which simpler diagrams cannot do. Additional information has been added to the legends to Figures 3, 6, 8 and 10 to facilitate understanding of the figures.

  1. Terminology such as "strain-specific stress response" needs to be defined clearly at the beginning. Furthermore, it is convenient to explain abbreviations, such as sDEGs, upon their first use in the Results section.

Answer: In the Introduction section, a clarifying phrase is given for the terminology used, "strain-specific stress response" (lines 95-96).

 The definition of "serotonergic differentially expressed genes (sDEGs)" is given upon their first use in the Results section (Line 125).

  1. Male rats only were used in this experiment. Note the sex limitation and whether these results could be applied to females.

Answer: Text has been added to the discussion section to draw attention to this limitation (lines 511-513).

  1. Reference [12] appears to report overlapping transcriptomic results. Can you please describe how this manuscript reports distinct conclusions from that work?

Answer: The Introduction section has been supplemented with text explaining the novelty of the results presented in this manuscript (lines 92-103).

  1. The identification of TFs like Egr1 and Tfap2a is intriguing. Are there any additional details or citations to prior studies that implicated these TFs in serotonergic regulation or hypertension?

 Answer: The Discussion section has been expanded according to the reviewer's recommendations (lines 462-503).

  1. A few of the conclusions, especially linking expression changes to functional cardiovascular consequences, seem speculative. Would you soften or support these comments with additional data or citations?

Answer: The phrase has been modified (Lines 517-520).

Comments on the Quality of English Language

The manuscript is generally readable and written in clear academic English. However, there are several instances where grammar, sentence structure, or word choice could be improved for clarity and flow.

Answer: The authors worked carefully with the text and tried to correct the shortcomings pointed out by the reviewer. The authors hope that the text has become completely clear and easy to read.

Round 2

Reviewer 2 Report

Comments and Suggestions for Authors

This revised manuscript is a thorough and well-organized examination of transcriptomic disparities associated with serotonergic signaling in the hypothalamus of ISIAH and WAG rat strains. The authors have carefully heeded previous reviewer feedback, providing extensive methodology, stating clearly the scope and novelty of their pathway-centric secondary analysis, and including augmented discussion of the major DEGs implicated in serotonergic and general neurotransmitter regulation under basal and stress conditions. The utilization of diverse bioinformatics software packages and databases, coupled with the clear visualization and functional annotations, greatly enriches the biological applicability and interpretability of the results. The manuscript now exhibits scientific stringency and offers useful insight into the genetic mechanisms of stress sensitivity and hypertension.

But one area that still needs to be improved is the figure legends, which are presently too concise and commonly lack enough context to lead the reader in comprehensively grasping the important observations and interpretations made in the figures. I suggest that the authors formulate longer figure legends to plainly outline the experimental comparison being illustrated, define any color codes or symbols used (e.g., red stars), state gene functions where applicable, and identify the main message of each figure.

With these small improvements, the manuscript will be ready for publication.

Author Response

The authors are grateful to the reviewer for carefully reading the manuscript and recommendations for its improvement. All corrections made to the text of the manuscript are shown in red.

Comments and Suggestions for Authors

This revised manuscript is a thorough and well-organized examination of transcriptomic disparities associated with serotonergic signaling in the hypothalamus of ISIAH and WAG rat strains. The authors have carefully heeded previous reviewer feedback, providing extensive methodology, stating clearly the scope and novelty of their pathway-centric secondary analysis, and including augmented discussion of the major DEGs implicated in serotonergic and general neurotransmitter regulation under basal and stress conditions. The utilization of diverse bioinformatics software packages and databases, coupled with the clear visualization and functional annotations, greatly enriches the biological applicability and interpretability of the results. The manuscript now exhibits scientific stringency and offers useful insight into the genetic mechanisms of stress sensitivity and hypertension.

But one area that still needs to be improved is the figure legends, which are presently too concise and commonly lack enough context to lead the reader in comprehensively grasping the important observations and interpretations made in the figures. I suggest that the authors formulate longer figure legends to plainly outline the experimental comparison being illustrated, define any color codes or symbols used (e.g., red stars), state gene functions where applicable, and identify the main message of each figure.

Answer: The authors have made corrections to the figure legends in accordance with the reviewer's recommendations.

With these small improvements, the manuscript will be ready for publication.
